# Enhancing Concrete and Mortar Properties and Durability Using Pristine Graphene Particles

Kirthi Chetty [1], Michael Watson [2], Thomas Raine [2], Todd McGurgan [3], Paul Ladislaus [2], Jun Chen [4], Shuai Zhang [4], Liangxu Lin [4] and Guangming Jiang [1],*

1 School of Civil, Mining and Environmental Engineering, University of Wollongong, Wollongong, NSW 2522, Australia
2 First Graphene (UK) Ltd., Graphene Engineering Innovation Centre, The University of Manchester, Manchester M13 9PL, UK
3 First Graphene Ltd., Henderson, WA 6166, Australia
4 Intelligent Polymer Research Institute, ARC Centre of Excellence for Electromaterials Science, University of Wollongong, Wollongong, NSW 2522, Australia
* Correspondence: gjiang@uow.edu.au or guangming.jiang@gmail.com

**Abstract:** The usage of industrially generated graphene was explored in this work, with an emphasis on dosage effects on durability, as well as the mechanical and microstructural properties of both concrete and mortar (0%, 0.1%, and 0.2% in concrete and 0%, 0.07%, and 0.15% in mortar). Based on the mix design for wastewater infrastructure, the results showed that adding graphene to both concrete and mortar enhanced 28-day compressive strength by 10%–20%, with the best admixture level being 0.02%–0.1%. Graphene reduced the AVPV of mortar by 11.7%, and concrete by 19.3% at the optimal dosages, likely by reducing the number or size of pores in the paste. The 0.2% and 0.15% graphene reinforced concrete and mortar showed significant sulfate resistance, by reducing 62% and 60% of extension respectively, after exposure to a sulfate solution for 16 weeks.

**Keywords:** graphene; concrete; mortar; compressive strength; slump; average volume of permeable voids (AVPV); sulfate; durability



## 1. Introduction

The great compressive strength and excellent shapeability made concrete a good structural material [1]. Low tensile and flexural strength, poor ductility, and a lack of resistance to crack formation, on the other hand, can lead to poor durability and expensive maintenance costs [2,3]. Corrosion is another major issue that adversely affects the durability of concrete [4,5]. The majority of Australia's largest cities are located along the coast, where the corrosion of steel reinforcement has a substantial impact on the long-term longevity of concrete structures. Because of the presence of capillary pores, gel pores, and weak cement-aggregate contact zones, concrete is an inherently porous composite. Chloride ions from saltwater or the marine environment can seep into concrete via these pore systems. When the chloride content in the reinforcing steel reaches a certain level, the corrosion of steel rebar begins. Another cause of concrete damage is sulfate attack [6]. Studies have reported that the sulfate attack in concrete mainly depends on its pore structure and the hydration products [7–9]. Sulfate salts, such as those based on sodium, calcium, magnesium, potassium, and ammonium, have been shown to be hazardous to concrete [10].

The use of low-penetrability concrete and the control of cracking are commonly used methods employed to obtain durable concrete. Other traditional methods for achieving long-lasting concrete include lowering the water/cement ratio and/or increasing the moist curing duration [11]. The use of fly ash, blast furnace slag, and glass-based fibres has grown more common in recent years [12–17]. These methods, however, have failed to improve concrete's physical qualities and durability [13,18]. Because chemical and/or mechanical

faults in the cement structure generate the majority of the concrete damage, nanoscale treatments are crucial for enhancing mechanical performance and adding new functionality.

Nanoengineering is currently being employed to identify new ways to improvise the performance of concrete. The usage of nanoparticles increased mechanical characteristics while also slowing crack propagation [19]. Two important mechanisms affecting the strength of cementitious materials were revealed by adding nano silica powder, $TiO_2$ nanoparticles, and precipitated $CaCO_3$ nanoparticles to the cement matrix [20]. To begin with, large-surface-area nanoparticles speed up the cement hydration which leads to more gel-like calcium silicate hydrate (CSH). Second, because of the small size, they can be utilised as a filler, resulting in a denser microstructure. Low aspect ratio nanoparticles, on the other hand, are unsuccessful at halting fracture propagation at the nanoscale, and hence cannot increase reinforcing efficiency [21].

Nanofibers and nanotubes have been used as additives in cementitious composites and have been shown to increase mechanical properties [22–26]. However, due to a lack of interfacial regions between the nanoparticles and the cementitious materials, these carbon compounds were unable to achieve a complete bonding with cementitious materials [27]. Although still at an early stage, freshly created nanomaterials show potential in developing enhanced cement-based materials [28]. One such nanomaterial is graphene, which is known as the strongest substance on the planet [29].

Graphene comes in a variety of forms, such as graphene oxide (GO), pristine graphene (PRG) etc., each with their own set of functions and qualities [30]. It has been revealed that adding graphene to cement enhances hydration and durability by making the C-S-H crystal matrix denser [31,32]. PRG contains fewer flaws, has better crystallinity, and has a higher conductivity than reduced GO, since it is created from graphite in a electrochemical cell by a straight forward exfoliation process [30]. Graphene nanoplatelets (GNPs), another form of graphene, have been demonstrated to increase the mechanical, durability, and sensing properties of cement composites [33–40]. When GNPs were added to cement mortar at a concentration of 2.5%, they reduced the water penetration depth by 64%, chloride diffusion coefficient by 70%, and chloride migration coefficient by 31% [41].

Because it is water-compatible and extremely dispersible, GO produced by the acid exfoliation of graphite is the most researched graphene in cement composites [30]. Wang et al. (2015) found that adding 0.05% GO to cement pastes boosted the compressive strength by 40.4% and flexural strength by 90.5% [42]. Other researchers corroborated this by reporting that GO improved the strength and hardness of cement composites [24,25,43,44]. The compressive strength of cement paste containing 0.06% GO grew by 58.5% after 28 days, while the flexural strength of the mix with 0.04% GO was raised by 67.1%, according to their findings. According to Sharma and Kothiyal (2015), the cement mortar's 28-day compressive strength was enhanced by 86.3% with the addition of 0.1% GO [45]. However, a lower crystalline and a higher defect content, in addition to the inferior mechanical characteristics compared to PRG and rGO, limit the usage of the GO material [30].

Despite their superior mechanical qualities, PRG and rGO are less appealing for use in cement composites because they are very hydrophobic [39,46]. Recent studies have shown that the ultrasonication of the mixture of surfactants with rGO or PRG enhanced the dispersion in an aqueous solution and greatly improved the cement material's mechanical characteristics [39,47]. A week after the inclusion of 0.05% PRG in the cement mortar, the compressive strength increased by 8% and the flexural strength by 24% [39]. With the inclusion of 0.1% PRG, the compressive strength of cement mortar was raised by 19.9% after 28 days [48]. With minimal information on the properties of the graphene materials used, these investigations simply looked at the variations in mechanical characteristics between a control mortar and a mix having a PRG of either 0.05% or 0.1% [49]. A further study with 0%, 2.5%, 5.0%, and 7.5% PRG dosages with 8μm exhibited that its inclusion can reduce water penetration depth considerably [41]. However, due to the high dosage levels, PRG had no effect on the mechanical properties, according to this study. The clustering and formation of multilayer PRG sheets obstructed the interaction of PRG and the cement matrix. A recent

study with PRG concentrations of 0%, 0.05%, 0.1%, 0.5%, and 1% validated that the cement mortar containing 0.05% PRG can raise the compressive and flexural strength by about 8.3% and 15.6% at 28 days, respectively. However, these strengths begin to deteriorate when the concentrations of PRG exceed 0.05% because of PRG agglomeration [50]. Although these experiments demonstrated a relationship between the qualities of PRG-enhanced cement mortars and PRG doses, the mechanisms remain unknown.

It has been reported that, as the smaller GO contains more functional groups, it often displays a stronger increase in mechanical properties than larger GO [18,45,49]. Unlike GO, PRG materials (e.g., graphene sheets) have fewer oxygen groups around the edges, which makes them unlikely to interact with the cement matrix and create friction and adhesion forces. As a result, it is hypothesised that the larger the size of a single PRG structure, the better the interaction with the cement matrix. While graphene materials are steadily making progress in terms of commercial adoption, with several graphene manufacturers across the world capable of producing industrially relevant quantities, there is still a lack of research on the use of industrially produced graphene to increase cementitious performance.

In order to fill the aforesaid research gaps, this research strives to investigate the use of graphene nanoplatelets with a focus on studying the dosage effects (0%–0.2% in concrete and 0%–0.15% in mortar) of small PRG ($D_V50 = 50$ μm) particles. Moreover, the cost of generating graphene in large amounts has significantly decreased because of industrial production. This is making it possible to include graphene into materials used on an industrial scale, such as concrete [2]. Thus, the primary aim is to use industrially manufactured graphene nanoplatelets to achieve enhanced durability (resistance to acid induced deterioration) in corrosive environments, such as sewers and to investigate the positive changes of mechanical and microstructural properties that support the enhanced corrosion resistance. The concrete and mortar properties, including mechanical strength, micropore structure, and durability, were systematically determined by measuring the slump, 28-day compressive strength, apparent volume of pore voids (AVPV), and sulfate resistance. The obtained results provide insights into the development of PRG enhanced concrete and mortar with high durability.

## 2. Materials and Methods

### 2.1. Graphene Particles and Dispersion Method

Most of the earlier investigations used graphene produced in the labs, with inconsistent traits and less repeatable structures and mechanical properties. Here, industrially produced graphene nanoplatelets powder (PureGRAPH® 50) from First Graphene Ltd. in Perth, Australia has been used. The estimated cost of graphene nanoplatelets used for this study is (USD) $20/kg. The physical characteristics of the graphene powder are presented in Table 1, Section S1 of Supplementary Materials.

**Table 1.** Physical characteristics of graphene.

| Product | Average Particle Size (μm) | Thickness (nm) | Purity (%) | Tapped Density (g/cm$^3$) |
|---|---|---|---|---|
| PureGRAPH® 50 | 50 | 16.7 | 92.6 ± 0.5 (XPS) <br> 89.7 ± 0.4 (Dumas method) | 0.3 |

For concrete and mortar coupon preparation, the PureGRAPH® powder was dispersed using a Branson 450 Digital Sonifier (Danbury, CT, USA). For concrete coupon preparation, PureGRAPH® 50, superplasticizer, and water were measured first in accordance with the mix design in Table 2. After adding the PureGRAPH® 50 and superplasticizer to water, the combination was sonicated for 75 min using the Branson Digital Sonifier (450 W, 20 kHz). For the mortar coupon preparation, PureGRAPH® 50 and water were first weighed, according to Table 3, and then the mixture of PureGRAPH® 50 and water was sonicated for 75 min. To avoid a rise in temperature, the probe sonication was carried out in pulse mode with the pulse on and off times set at 1 s. The sonicated mix was immediately used for the cast of concrete or mortar coupons. The particle size of graphene before/after

sonication was also analysed using Malvern Mastersizer to check if the sonication reduced the apparent particle size.

**Table 2.** Mix design using PureGRAPH admixture for new concrete structures.

| | Constituents [a], (kg/m$^3$) | | | | | | | Admixture | |
|---|---|---|---|---|---|---|---|---|---|
| | w/c [b] | Cement | Water | Aggregates [c] | | | | PureGRAPH® 50 | S.P. [d] |
| | | | | 10 mm | MS | CS | FS | | |
| Control | 0.4 | 420 | 168 | 750 | 375 | 469 | 281 | 0 | 4.2 |
| PureGRAPH® 50-Low | 0.4 | 420 | 168 | 750 | 375 | 469 | 281 | 0.10% | 4.2 |
| PureGRAPH® 50-High | 0.4 | 420 | 168 | 750 | 375 | 469 | 281 | 0.20% | 4.2 |

[a] The mass required to form 1 m$^3$ of concrete is given for each constituent. [b] w/c: water-to-cement mass ratio. [c] 10 mm—crushed aggregates, MS—manufactured sand, CS—coarse sand, FS—natural fine sand. [d] S.P.: superplasticizer.

**Table 3.** PureGRAPH repair mortar for existing damaged structures.

| | Constituents [a], (kg/m$^3$) | | | | | Admixture |
|---|---|---|---|---|---|---|
| Samples | w/c [b] | Cement | Water | CS [c] | FS [d] | PureGRAPH® 50 |
| Control (no treatment) | 0.5 | 600 | 300 | 863 | 464 | 0 |
| PureGRAPH® 50-Low | 0.5 | 600 | 300 | 863 | 464 | 0.07% |
| PureGRAPH® 50-High | 0.5 | 600 | 300 | 863 | 464 | 0.15% |

[a] The mass required to form 1 m$^3$ of concrete is given for each constituent. [b] w/c: water-to-cement mass ratio. [c] CS: coarse sand. [d] FS: fine sand.

## 2.2. Preparation of Concrete and Mortar Coupons

The mix design for concrete and mortar coupons is given in Tables 2 and 3. The binder for both the mortar and the concrete mix was Ordinary Portland Cement (OPC) [51]. MasterGlenium SKY 8100 that meets AS 1478.1-2000 was utilised as a surfactant in concrete mixes to improve the graphene dispersion and workability [52]. For concrete, PureGRAPH® 50 was added at 0.1% and 0.2% of cement by weight, whereas for mortar it was added at 0.07% and 0.15%.

To prepare concrete coupons, the coarse aggregate, along with coarse and fine sand, were charged into the mixer, which was run for 30 sec, followed by cement addition, and further mixing for 2 min. The probe sonicated mix with water, superplasticizer, and PureGRAPH® 50 was then added and mixed for 2 min (Section 2.1). For the mortar coupon preparation, the coarse sand and fine sand were first added to the bowl and stirred for 1 min. The cement was then added and again blended for 3 min. The probe sonicated mixture of water and PureGRAPH® 50 (Section 2.1) was then added to the above prepared mixture, and then was mixed for 5 min. Before placing the concrete and mortar mix into their respective moulds, a thin coat of releasing agent WD-40 silicone lube was sprayed on to the inner surface of the moulds. Compaction by vibrating and rodding was carried out for concrete and mortar, respectively, after filling the moulds for uniform distribution over the cross-section of the mould. All of the moulds were let to cure for 24 h at ambient temperature. Later, demoulding was carried out and they were cured for 28 days in a solution of lime and water [53]. Different concrete and mortar coupons were prepared for different mechanical and durability tests. The test details with curing periods are given in the following sections.

## 2.3. Mechanical Property and Durability Tests

### 2.3.1. Slump or Workability Test

Workability is attributed to a product's capability to be combined, managed, carried, and installed with minimal homogeneity loss, such as segregationor bleeding. The workability of the new concrete and mortar was accomplished in line with AS 1012.3.1 [54]. The slump cone test was employed to assess the slump value, which entailed pouring

fresh concrete and mortar in four layers, each 1/4th of the height of the cone. Later the layers were tamped 25 times prior to lifting and calculating the height. Slump values were recorded separately for the various concrete and mortar mixtures.

### 2.3.2. 28-Day Compressive Strength and Density

Compressive strength tests were executed on the concrete and mortar in line with AS 1012.9 and AS 2350.11, respectively, at an age of 28 days using a 1800 kN Avery Compression Testing Machine [55,56]. The details of the admixture, dimensions, and number of samples for the test are given in Tables S2 and S3 of the Supplementary Materials. A force of 0.333 Mpa/s was enforced until no more force could be tolerated and the highest force applied was registered. For the concrete and mortar coupons, the density was measured according to AS 1012.12.1. [57]. Both of these tests were conducted in triplicates.

### 2.3.3. Apparent Volume of Permeable Voids (AVPV)

The determination of *AVPV* in both hardened concrete and mortar was carried out in line with AS 1012.21 [58]. Test specimens were Ø100 × 200 mm cylinders and were sliced into four equal slices that were trimmed to a maximum of 3 mm prior to slicing. For the immersed absorption ($A_i$), to the nearest 0.1 g, the sample was weighed using a weighing balance (2200 g Scout General Portable Balance) and later was left in the oven (Thermoline Scientific Bench top—TO-SERIES) at 100 to 110 °C for at least 24 h to dry, with special care taken to avoid touching with other specimens. After drying in the oven, the samples were cooled to 23 ± 2 °C in a desiccator prior to weighing. If the difference between two consecutive measurements was larger than 1 g, the specimen was returned to the oven for another 24 h of drying, and the process was repeated until the difference between two consecutive readings was less than 1 g, at which point the mass, $M_1$, was recorded. The sample was submerged in distilled water at room temperature for at least 48 h, and then the weight was measured every 24 h. If the weight increased by less than 1 g after 48 h, the mass $M_{2i}$, was recorded. The saturated specimen was surface-dried by removing the surface moisture with a towel. For boiled absorption ($A_b$) and *AVPV*, the surface-dried specimen was placed in a room-temperature water bath, ensuring that it was completely submerged. It was then boiled for 5.5 ± 0.5 h., then allowed to cool naturally for at least 14 h while still immersed in the water bath to reach the room temperature. The surface-dried and boiled specimen was then weighed, and the mass, $M_{3b}$, was recorded. After that, the specimen was immersed in water by suspending it on a rack at a temperature of 23 ± 2 °C, with the mass, $M_{4ib}$, being recorded.

The results for *Ai*, *Ab*, and *AVPV* were calculated using the following equations:

$$A_i = \frac{(M_{2i} - M_1)}{M_1} \times 100\% \tag{1}$$

$$A_b = \frac{(M_{3b} - M_1)}{M_1} \times 100\% \tag{2}$$

$$AVPV = \frac{(M_{3b} - M_1)}{M_{3b} - M_{4ib}} \times 100\% \tag{3}$$

### 2.4. Sulfate Resistance Testing

The assessment of sulfate resistance on the concrete and mortar samples with and without the PureGRAPH® 50 admixture was conducted in accordance with AS 2350.14 [59]. Because of the presence of 10 mm aggregates in the concrete mix, prismatic moulds of dimensions 40 × 40 ×150 mm were used. For the mortar, prismatic moulds of 40 × 15 × 150 mm were used. These samples were cured for 7 days, and then were kept at room temperature for 1 complete day. Later they were immersed in a standard sulfate solution (0.352 mol/L of $Na_2SO_4$). The lengths of the specimens were initially measured after 7 days of curing and subsequently every 2 weeks and up to 16 weeks after immersing in the sulfate solution.

The sulfate solution was replaced for every 2 weeks, and the pH was maintained with the help of 0.05 mol/L (0.1 N) sulfuric acid between 6 and 8 at regular periods. On the day of analysis, the specimens were removed from the solution, wiped, and the length change was measured with a length comparator (resolution of 0.001 mm). The sulfate testing was carried out in triplicates for both concrete and mortar samples.

## 3. Results and Discussion

### 3.1. PureGRAPH® 50 Dispersion in Water

The particle size of PureGRAPH® 50 was measured prior to and at 30 min, 45 min, 60 min, and 75 min of the probe sonication. The results indicated that the particle size of PureGRAPH® 50 did not change, even after 75 min of probe sonication and remaining at 56.3 µm (D50) (Figure S1 and Table S4). The probe sonication dispersed the PureGRAPH® 50 in water in the case of mortar preparation and when the superplasticizer was added for concrete preparation, the mixture turned thick (Figure S2). After sonication, the PureGRAPH® 50/water mixture started to rise with time while, in the presence of superplasticizer, the dispersion was prominent and persistent. The dispersion studies of Kaur et al. (2020) showed that the superplasticizer improved the dispersion ability for a longer time [60]. The superplasticizer has a "comb-like" molecular structure consisting of an anionic adsorbing main chain and non-adsorbed non-ionic side chain [61,62]. Non-ionic surfactants can be employed to successfully disperse graphene in water at high concentrations. Anionic surfactants have been proven to be efficient at dispersing and stabilizing graphene particles in aqueous solutions, even at lower concentrations. Thus, the combined action of non-ionic and anionic components in the superplasticizer supports the ability to disperse the graphene in aqueous solution [47,62].

### 3.2. Workability—Slump Test

The results obtained from the slump test of concrete and mortar are shown in Figure 1, Figures S3 and S4. The slump obtained for concrete samples with 0.1% and 0.2% PureGRAPH® 50 (all following percentages for graphene and other admixtures are inexplicitly in the weight % of cement) are lower than that of the control by 79.5% and 84%, respectively. The same decreasing trend in slump (when compared to the control) was also seen for mortar samples. The 0.07% and 0.15% PureGRAPH® 50 admixed mortar showed slump values of 22.2% and 77.7%, respectively, which were lower than the control. Collectively speaking, with the increment in the dosage of the graphene, the slump values decreased. Similar findings were outlined by Lu et al. (2017) who investigated concrete's ultra-high strength with GO nanosheets [63]. The workability of cement composites with PureGRAPH® 50 is shown to steadily decline as the graphene dosage in the cement composites increased in most prior investigations. Graphite's large specific area, as well as agglomerated graphene, could potentially block the migration of fresh composites by trapping water inside the flocs [39,64,65]. Jiang et al. (2017)s discovered that, when graphene was added to cement paste at 0.2%, the paste's flowability decreased by 17.4%, and when the concentration was 0.4%, the flowability decreased by 39% [66].

The workability of graphene assisted composites can be improved by raising the surfactant amount and the ultrasonication treatment period. When the dosage of the superplasticizer is increased to 0.6%, 0.9%, 1.2%, and 1.5%, and the graphene concentration is at 1%, 1.5%, 2%, and 2.5%, Du et al. (2016) showed that graphene reinforced concrete can flow just like conventional concrete [17]. They also tested the flowability with different superplasticizer dosages but with the same graphene content. They discovered that adding 1% graphene to cement paste greatly affects its workability, whereas increasing the superplasticizer dosage effectively increases the workability of the graphene-infused cement paste [64].

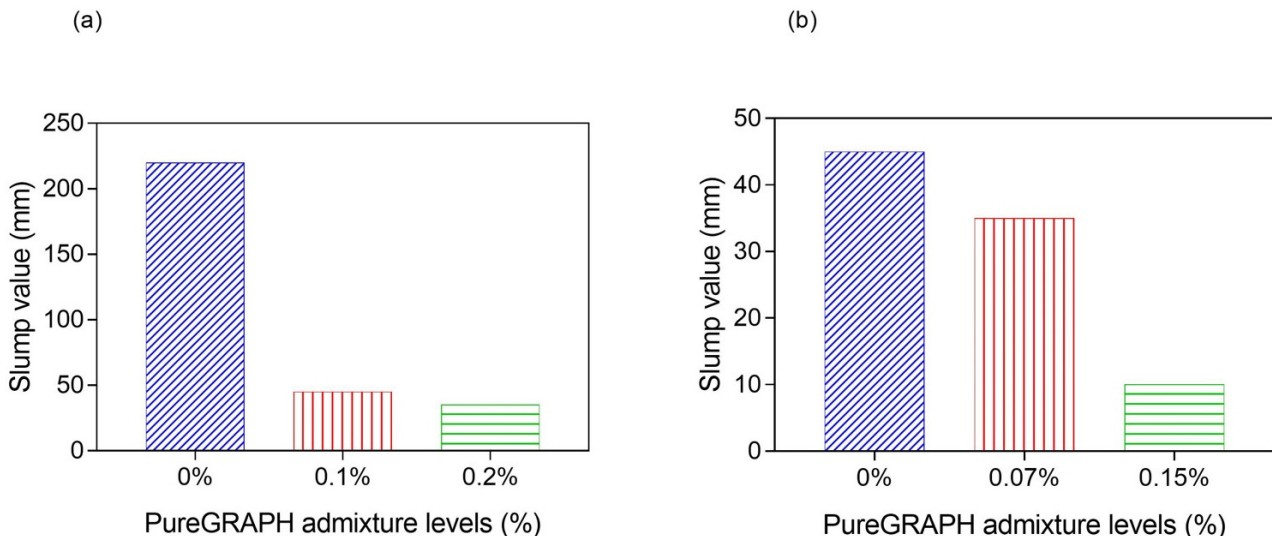

**Figure 1.** Slump of (**a**) concrete and (**b**) mortar with different weight % of PureGRAPH® 50.

It is worth mentioning that previous research found that increasing the amount of graphene had no influence on the workability of graphene reinforced cement composites [67,68]. The very low concentration of graphene or the graphene with a reduced surface area that require a lower amount of water to saturate its surface might have affected the flowability [67].

### 3.3. 28-Day Compressive Strength and Density

The compressive strength tests on PureGRAPH® 50 reinforced concrete and mortar were conducted on a compressive strength test machine after 28-days of a curing period, with the results given in Figure 2 and Table S5. Compressive strengths were higher in the PureGRAPH® 50 mixtures than in the control. For concrete samples, the 0.1% and 0.2% PureGRAPH® 50 samples showed a 9.9% and 7.4% increase, respectively, in the compressive strength, in comparison to the control. The same trend was also observed for mortar samples. The 0.07% and 0.15% PureGRAPH® 50 mortar samples gave a 19.8% and 7.5% increase, respectively, in comparison to the control mortar sample. As the PureGRAPH® 50 dosage increased above 0.1% in both concrete and mortar samples, the compressive strength slightly decreased when compared to the samples with less than 0.1%. The statistical analysis showed that the optimum PureGRAPH® 50 concentration must be ≤0.1% (95.6% confidence level) to meet the 50 MPa requirement.

Considering previous research, the majority of the data depicted that the best mechanical properties of graphene-admixed cement composites are obtained with graphene dosages ranging from 0.02% to 1%. Qureshi et al. (2020) compared the mechanical characteristics of cement pastes, including 0.01% to 0.16% graphene. They discovered that 0.02% graphene has the highest compressive and flexural strength of the cement paste, but if this amount is exceeded, the compressive and flexural strength started to decrease [69]. Wang et al. (2019) investigated the mechanical properties of cement composites admixed with 0.03%, 0.06%, and 0.09% graphene, and found that 0.06% graphene loading had the highest performance. However, increasing the graphene content further reduced the compressive and flexural strengths [70]. These above conclusions made by Wang et al. (2019) for the optimal dosage of graphene are also quite similar to Madbouly et al. (2020) [71]. Surprisingly, Du et al. (2016) reported slightsly different results, claiming that introducng 1% graphene results in concrete with the maximum compressive strength [17].

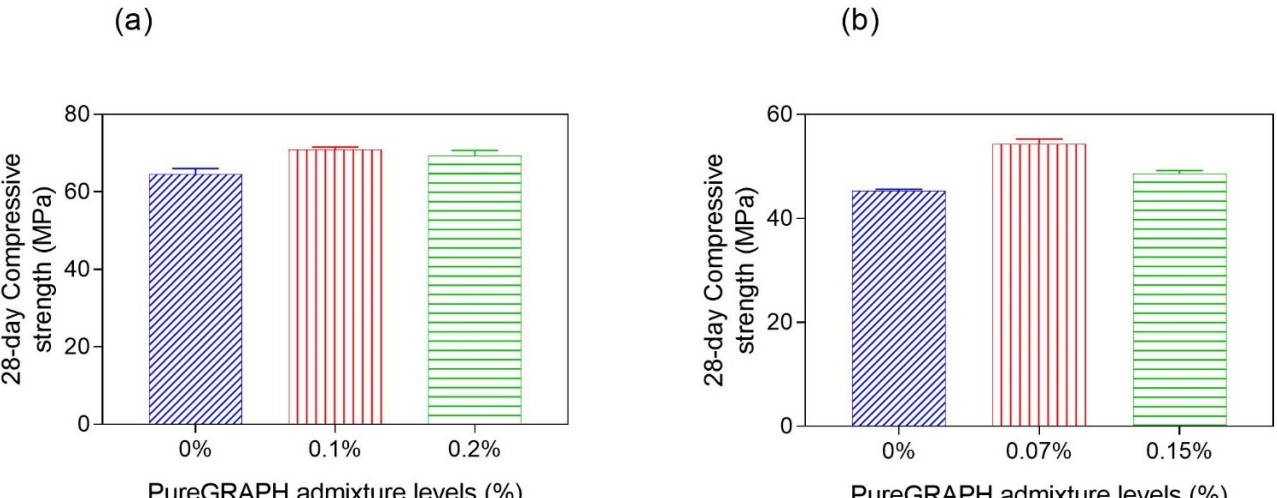

**Figure 2.** Compressive strength of PureGRAPH® 50 reinforced (**a**) concrete and (**b**) mortar samples at different levels. The standard deviation of the triplicate measurements is indicated by the error bars.

Different processing conditions, such as the type of dispersant (water with/without superplasticizer), the time span of ultrasonication, the graphene platelet size, and the sonicator amplitude, could explain the differences in optimal graphene dosages on the mechanical characteristics of cement composites compared to prior investigations. As a result, determining an exact figure for the appropriate graphene dose to produce the highest mechanical characteristics in cement composites is difficult. The best dose of graphene for improving the mechanical characteristics of cement composites appears to be 0.02%–0.1%. A graphene dosage of more than 1% is considered excessive because graphene aggregation has an adverse effect on the mechanical characteristics of cement composites.

The density of graphene admixed concrete and mortar samples is given in Table S6. For admixed concrete samples, the density of 0.1% and 0.2% samples increased by 2.0% and 3.1%, respectively, when compared to the control, which had a density of $2343.2 \pm 0.4 \ \mathrm{kg/m^3}$. For the mortar samples, the density increased by 2.4% and 2.8% for the 0.07% and 0.15% admixed mortar samples, respectively, compared to the control, which had a density of $2078.01 \pm 0.1 \ \mathrm{kg/m^3}$. A study conducted by Long et al. (2017) using the GO nano-scale layer material proved that the GO make the material more solid by filling all of the pores [72]. It is worth noting that, if the matrix is denser, the material's mechanical characteristics will be improved [73].

*3.4. Apparent Volume of Permeable Voids*

Figure 3 and Table S7 depict the *AVPV* of the PureGRAPH® 50 admixed concrete and mortar samples. The *AVPV* for the control concrete and mortar is 11.1% and 12.9%, respectively. The addition of PureGRAPH® 50 at a concentration of 0.1% decreased the *AVPV* by 11.7%, and, for a 0.2% concentration, the *AVPV* reduced by 7.2%, when compared to the control concrete. For the graphene admixed mortar samples, the *AVPV* decreased by 19.3% and 12.4%, respectively, when 0.07% and 0.15% of graphene was added. The addition of graphene has reduced the absorption and volume of the permeable voids. However, the relationship between graphene content, immersed absorption ($A_i$), and the *AVPV* is nonlinear, with 0.07% and 0.1% resulting in the lowest absorption and *AVPV* of mortar and concrete samples, respectively. This is consistent with graphene influencing the microstructure of the cement paste and concrete, perhaps lowering the number and size of pores in the cement paste.

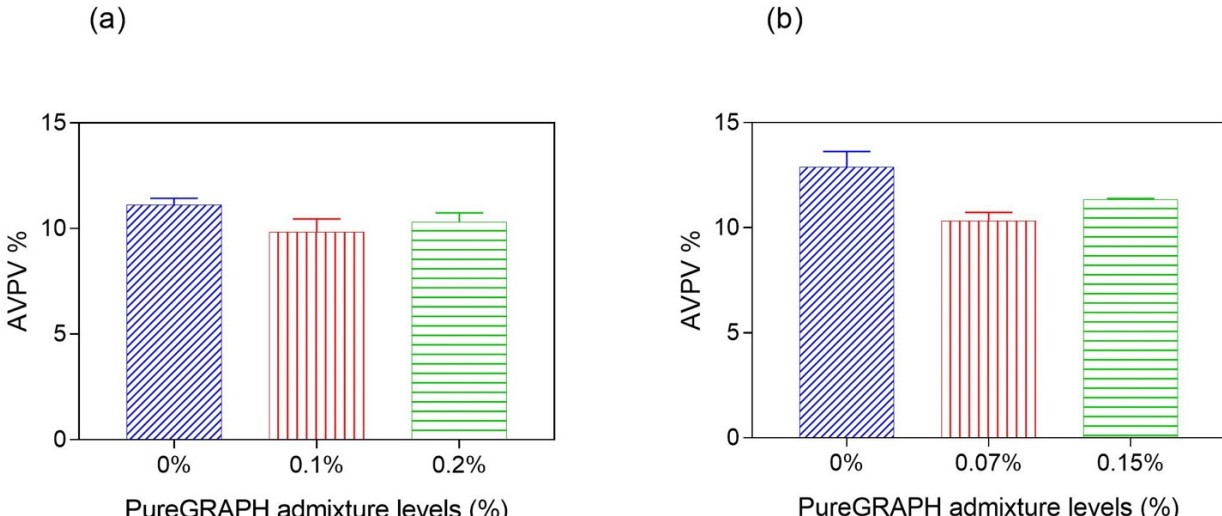

**Figure 3.** AVPV of the PureGRAPH® 50 reinforced (**a**) concrete and (**b**) mortar samples at different admixture levels. The standard deviation of the triplicate measurements is indicated by the error bars.

With regards to water absorption, there was little difference between the immersed ($A_i$) and boiled ($A_b$) absorption, suggesting that all permeable voids within the concrete can be accessed by immersion in water only. The reduced *AVPV* is expected to have a beneficial impact on concrete durability, as the addition of graphene changes the cement paste or concrete microstructure, resulting in a decrease in porosity and, therefore, subsequent absorption of aggressive species. In general, a reduced-porosity concrete which contains fewer permeable voids is less likely to degrade because of the increased difficulty with which sulfate, chlorides, and carbonic acid (in the form of $CO_2$ gas) can ingress into the concrete. This is particularly true of reinforced concrete, which requires a level of concrete cover to protect it from corrosion.

*3.5. Sulfate Resistance Testing*

The data showing the effect of sulfate attack on the PureGRAPH® 50 admixed concrete and mortar samples are shown in Figure 4 and Table S8. The control concrete increased by 1.5 mm in length, whereas the control mortar sample had a length increase of 1.8 mm on week 6, with a slight increase afterwards until week 16. In comparison, the extension of graphene admixed concrete only reached to 0.8 mm and 0.5 mm for 0.1% and 0.2% admixture levels, respectively, on week 6. The extension reached 2.6 mm and 0.7 mm for 0.07% and 0.15% graphene admixed mortar samples, respectively, after week 6. The expansion after week 6 until week 16 was very slow in the case of both graphene admixed concrete and mortar samples. Given the ability to modify the pore structure, improve the strength, and reduce the permeability, concrete and mortar containing a high percentage (0.2%–0.15%) of graphene particles showed superior sulfate resistance compared to the control.

Sulfate attack is the most prevalent sort of chemical attack that lowers concrete durability in real-world applications [74–76]. Sulfate ions can be found in groundwater, clayey soils, and areas near heavy industry and their associated effluent. Sulfate is also found in seawater, polluted rainfall, and at a high level in sewage [5,77]. Concrete can crack and expand as a result of sulfate attack, mainly because of the development of gypsum and ettringite [2,78–80]. The sulfate induced expansion was found to decrease consistently with a pristine graphene partsicle content of 0.07% or more for both types of samples. Sulfate ions may preferentially bind to graphene, rendering them ineffective on the quality of hydrates [81]. The high sulfate resistance may also be linked to the reduced AVPV values and, subsequently, the lower chemical diffusion rate in the mortar or concrete samples.

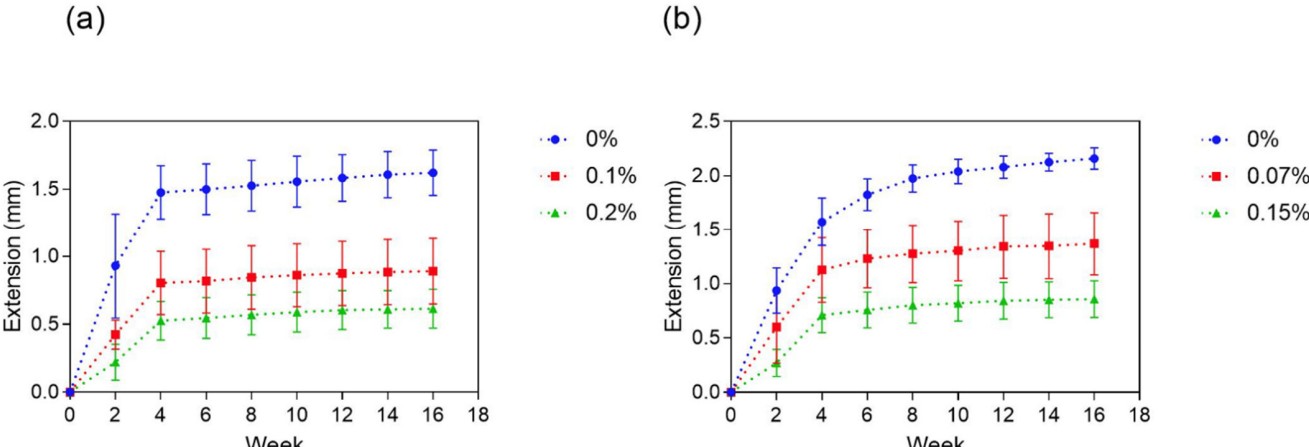

**Figure 4.** Sulfate resistance of the PureGRAPH® 50 reinforced (**a**) concrete and (**b**) mortar samples at different ad-mixture levels. The standard deviation of the triplicate measurements is indicated by the error bars.

## 4. Conclusions

This study investigated the enhanced mechanical properties and durability of concrete and mortar composites with industry produced PureGRAPH® 50 powder as an admixture at different levels, through measuring the slump, 28-day compressive strength, density, AVPV, and sulfate resistance. From the current investigation, the following conclusions can be drawn:

PureGRAPH® 50 enhanced the 28-day compressive strength of both concrete and mortar, with the optimal percentage being in the range of 0.02%–0.1%, while workability was lowered, which might be addressed by using superplasticiser.

The addition of PureGRAPH® 50 reduced the AVPV, potentially by decreasing the number or pore sizes in the cement paste. The PureGRAPH® 50 concentration, water absorption, and AVPV have a nonlinear connection, with the lowest absorption and AVPV in mortar and concrete samples (0.07% and 0.1%), respectively.

When compared to control samples, PureGRAPH® 50 reinforced concrete and mortar demonstrated dramatically improved sulfate resistance. This improved sulfate resistance has the potential to considerably improve concrete's corrosive environment durability. Thus, the reinforced concrete with graphene nanoplates meet the high requirement of durability for many new infrastructures to ensure adequate service life.

**Supplementary Materials:** The following supporting information can be downloaded at: https://www.mdpi.com/article/10.3390/coatings12111703/s1.

**Author Contributions:** Conceptualization, M.W. and G.J.; methodology, K.C., T.R., J.C., S.Z., L.L. and G.J.; Experimental design, G.J.; investigation, K.C.; resources, G.J.; writing—original draft preparation, K.C.; writing—review and editing, T.M., P.L. and G.J.; visualization, K.C.; supervision, G.J.; All authors have read and agreed to the published version of the manuscript.

**Funding:** This research was funded by University of Wollongong RevITAlising (RITA) Research Grants (2021).

**Institutional Review Board Statement:** Not applicable.

**Informed Consent Statement:** Not applicable.

**Data Availability Statement:** Not applicable.

**Acknowledgments:** Kirthi Chetty receives the support from a University of Wollongong scholarship.

**Conflicts of Interest:** The authors declare no conflict of interest.

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
