# Peer review of "Enhancing Concrete and Mortar Properties and Durability Using Pristine Graphene Particles"

_coatings, doi:10.3390/coatings12111703_

Round 1
Reviewer 1 Report
In this work, Chetty et al. present a systematic study for the enhancement of Concrete and Mortar Properties by using Pristine Graphene. Unfortunately, the innovation of the work is not apparent, as no significant improvement in the mechanical properties of Concrete or Mortar with respect to other published works is clearly addressed. Also, there are other major flows that need to be improved before its publication.
1) The references are not properly added from page 3 till the end of the paper.
2) The quality of the figures is really low
3) The authors use the term PRG for pristine Graphene for the compound Pure GRAPHH 50. Unfortunately, it is not graphene. Graphene corresponds to a single layer of graphite. As the data in table 1 indicates the thickness of the compound is 16.7 nm. Considering that a single layer of graphite is 0.6 nm thick, the compound they are using is a multilayer, therefore exfoliated graphite, no graphene. The authors should consider this and change the word graphene to exfoliated graphite.
4) It is not clear how the authors determine the parameters in table 1. Authors should indicate the techniques used and present the results in the Supporting material.
5) The authors claim that the Pure GRAPHH 50 is redispersed in water after tip sonication. Pure Graphene ( or exfoliated graphite, which is what the authors have) is hydrophobic, it cannot be properly stabilized in aqueous solutions without the use of a surfactant. The authors should explain this, how it is possible? Authors should study XPS and Raman, the quality of the Pure GRAPHH 50 after the sonication, it is possible that during the treatment, the surface of the flakes got oxidized. Therefore, having O and OH groups that may affect the physical properties of concrete and mortar.
Author Response
Dear Editor,
We are grateful for the constructive comments received from the reviewers, which helped us to further improve the quality and clarity of the manuscript. We appreciate the opportunity to revise this manuscript and have carefully evaluated and addressed all the comments and amended the manuscript accordingly. Manuscript ID: coatings-1996686 Below are our detailed responses to the reviewer’s comments point by point. The comments from the editor and reviewers are in black, responses from the authors are in blue, and revisions to the manuscript are in red. We would be happy to address any further comments that you or the reviewers might have.
Kind regards,
Dr. Guangming Jiang
School of Civil, Mining and Environmental Engineering,
University of Wollongong, Australia.
On behalf of all the authors
COMMENTS FROM THE EDITOR AND/OR REVIEWERS
Your manuscript has now been reviewed by experts in the field. Please find your manuscript with the referee reports. Please revise the manuscript according to the referees’ comments.
We appreciate all the constructive and insightful comments on the manuscript made by all reviewers and the editor. We carefully revised the manuscript by taking all these comments into consideration, as shown below.
Reviewer 1:
In this work, Chetty et al. present a systematic study for the enhancement of Concrete and Mortar Properties by using Pristine Graphene. Unfortunately, the innovation of the work is not apparent, as no significant improvement in the mechanical properties of Concrete or Mortar with respect to other published works is clearly addressed. Also, there are other major flows that need to be improved before its publication.
We agree with the reviewer that there are many other studies being done using various graphene products to enhance concrete or mortar properties. These studies were reviewed and discussed in the introduction section. While graphene materials are steadily making progress in terms of commercial adoption, with several graphene manufacturers across the world capable of producing industrially relevant quantities, there is still a lack of research on the use of industrially produced graphene to increase cementitious performance are still lacking. Therefore, this study systematically assessed the use of of industrially manufactured graphene with a focus on studying the dosage effects (0%-0.2% in concrete and 0%-0.15% in mortar) of small PRG (DV50 = 50 µm) particles on boosting the durability together with the mechanical and microstructural characteristics of both concrete and mortar. This study is systematic by integrating mechanical, microstructural and durability measurements, which are generally not reported in other studies. Especially, our research group mainly focus on the corrosion resistance and durability in urban wastewater systems. Therefore, this study is mainly designed to achieve good durability but also to investigate the supporting mechanisms by investigating the microstructural changes induced by the use of PRG. We have made the following changes to the manuscript to clearly distinguish this study from other published works.
Line 194:
Thus, the primary aim is to use industrially manufactured graphene nanoplatelets to achieve enhanced durability (resistance to acid induced deterioration) in corrosive environment like sewers and to investigate the positive changes of mechanical and microstructural properties that support the enhanced corrosion resistance.
- The references are not properly added from page 3 till the end of the paper.
The references have been properly added in the manuscript.
- The quality of the figures is really low.
The figures are now properly submitted to the manuscript with high quality.
Figure 1. Slump of (a) concrete and (b) mortar with different weight % of PureGRAPH® 50
Figure 2. Compressive strength of PureGRAPH® 50 reinforced (a) concrete and (b) mortar samples at different levels. The standard deviation of the triplicate measurements is indicated by the error bars.
Figure 3. AVPV of the PureGRAPH® 50 reinforced (a) concrete and (b) mortar samples at different admixture levels. The standard deviation of the triplicate measurements is indicated by the error bars.
Figure 4. Sulfate resistance of the PureGRAPH® 50 reinforced (a) concrete and (b) mortar samples at different ad-mixture levels. The standard deviation of the triplicate measurements is indicated by the error bars.
3) The authors use the term PRG for pristine Graphene for the compound Pure GRAPHH 50. Unfortunately, it is not graphene. Graphene corresponds to a single layer of graphite. As the data in table 1 indicates the thickness of the compound is 16.7 nm. Considering that a single layer of graphite is 0.6 nm thick, the compound they are using is a multilayer, therefore exfoliated graphite, no graphene. The authors should consider this and change the word graphene to exfoliated graphite.
Thank you for the suggestion. “Graphene nanoplatelets” suits for the material we have used. So, the graphene word has been replaced with graphene nanoplatelets and PureGRAPH® 50 wherever applicable to clear the confusion.
Line 206:
Here, industrially produced graphene nanoplatelets powder (PureGRAPH® 50) from First Graphene Ltd in Perth, Australia has been used.
Line 224:
For concrete coupon preparation, PureGRAPH® 50 , superplasticizer, and water were measured first in accordance with the mix design in Table 2. After adding the PureGRAPH® 50 and superplasticizer to water, the combination was sonicated for 75 min using the Branson Digital sonifier (450W, 20kHz). For mortar coupon preparation, PureGRAPH® 50 and water were first weighed according to Table 3 and then the mixture of PureGRAPH® 50 and water was sonicated for 75 min.
Line 240:
For concrete, PureGRAPH® 50 was added at 0.1% and 0.2% of cement by weight whereas for mortar it was added at 0.07% and 0.15%.
Line 287:
The probe sonicated mix with water, superplasticizer and PureGRAPH® 50
Line 290:
The probe sonicated mixture of water and PureGRAPH® 50 (Section 2.1)
Line 349:
The assessment of sulfate resistance on the concrete and mortar samples with and without PureGRAPH® 50 admixture was conducted in accordance with AS2350.14
Line 367:
The probe sonication dispersed the PureGRAPH® 50 in water in case of mortar preparation and when superplasticizer was added for concrete preparation, the mixture turned thick (Figure SI-2). After sonication, the PureGRAPH® 50 / water mixture started to rise with time while in the presence of superplasticizer, the dispersion was prominent and persistent.
Line 391:
The workability of cement composites with PureGRAPH® 50 is shown to steadily decline as the graphene dosage in the cement composites increased in most prior investigations.
Line 416:
Figure 1. Slump of (a) concrete and (b) mortar with different weight % of PureGRAPH® 50.
Line 432:
The compressive strength tests on PureGRAPH® 50 reinforced concrete and mortar
Line 452:
Figure 2. Compressive strength of PureGRAPH® 50 reinforced (a) concrete and (b) mortar samples at different levels. The standard deviation of the triplicate measurements is indicated by the error bars.
Line 494:
Figure 3 and Table SI-7 depict the AVPV of the PureGRAPH® 50 admixed concrete and mortar samples.
Line 507:
Figure 3. AVPV of the PureGRAPH® 50 reinforced (a) concrete and (b) mortar samples at different ad-mixture levels. The standard deviation of the triplicate measurements is indicated by the error bars.
Line 520:
The data showing the effect of sulfate attack on the PureGRAPH® 50 admixed concrete and mortar samples are shown in Figure 4 and Table SI-8.
Line 553:
Figure 4. Sulfate resistance of the PureGRAPH® 50 reinforced (a) concrete and (b) mortar samples at different ad-mixture levels. The standard deviation of the triplicate measurements is indicated by the error bars.
Line 568:
This study investigated the enhanced mechanical and durability of concrete and mortar composites with industry produced PureGRAPH® 50 powder as an admixture.
Line 577:
The PureGRAPH® 50 concentration, water absorption, and AVPV have a nonlinear connection, with the lowest absorption and AVPV in mortar and concrete samples (0.07 % and 0.1%) respectively.
Line 604:
When compared to control samples, PureGRAPH® 50 reinforced concrete and mortar demonstrated dramatically improved sulfate resistance.
4) It is not clear how the authors determine the parameters in table 1. Authors should indicate the techniques used and present the results in the Supporting material.
The techniques used are presented in the Supporting material.
Section SI-1
PureGRAPH® 50 average particle size:
Using a Malvern Mastersizer 3000E, the average particle size of PureGRAPH® 50 was measured.
PureGRAPH® 50 thickness measurement:
PureGRAPH® 50 AQUA was combined with IPA/water mixture to produce solutions at 0.5 wt% (taking residual water into account). Initially the material was dispersed by manually shaking to break up larger agglomerates of wet cake. Dispersion was achieved by ultrasonic bath sonication (400 W) for 5 mins. Solutions were further diluted at 1:50 in fresh IPA/water. Silicon wafers (10 x 10 mm) were cleaned by bath sonication (400 W) for 5 mins in acetone, DI water and IPA (separately and transferring without drying). Samples were dried using CDA line blow-off gun.
Silicon wafers were spray coated with the PG50 solution using MTI spray pyrolysis chamber with ultrasonic head.
AFM images were acquired using JPK NanoWizard (housed at NGI) in intermittent contact mode. TESPA-V2 probes (Bruker) were used, which were made from Sb-doped silicon with a reflective aluminium coating on the back of the cantilever. Fast scans were first obtained to map out a 60 x 60 μm area on each sample. Each fast scan image was 20 x 20 μm and was acquired at a 0.5 Hz scan rate with a 128 px resolution. Slow scans were then used to map a 50 x 50 μm region for analysis. Each slow scan image was 25 x 25 μm and was acquired at a 0.2 Hz scan rate with a 256 px resolution.
The AFM flakes were analysed in Gwyddion software. The flake lateral size and thickness are usually loosely correlated, although it is hard to draw conclusions about the distribution in each sample as the sample set is too small. Sample size does not affect the conclusions drawn regarding contrast to height correlation, but the results would be refined with further data. The flakes were identified and labelled with an ID number. Each flake was masked and three parallel line profiles were drawn over the flake. Height of the flakes was measured as the difference between the planar regions of the flake and the height of the neighbouring substrate. Lateral dimensions were extracted by two perpendicular lines across the width and length of the flake.
Purity measurement: XPS
XPS method:
Samples were dispersed in ethanol (1mg/mL) and drop casted on a silicon substrate. Before analysis, the films were heated under vacuum at 60⁰C to remove any residual solvent prior to XPS measurements.
The X-Ray Photoelectron Spectroscopy (XPS) instrument was provided by SPECS (Berlin). A no-monochromatic X-ray source (12kV-200 W) with magnesium anode was used for the measurements. The operation was performed under ultra-high vacuum (UHV) condition with a base pressure of e-10 mbar.
The samples were mounted on a molybdenum sample holder. Semiconductor-grade silicon was used as a substrate. The conductivity of molybdenum holder and silicon substrate was sufficient for the electron compensation due to the X-ray radiation and thus avoiding any charging of the samples.
The fittings of the spectra were done using the software CasaXPS, by the Gaussian-Lorentzian functions with a Shirley background subtraction.
Dumas method:
Samples weighed precisely into tin capsules and dropped at pre-set times into combustion tube (at 1000°C). A constant stream of helium is maintained through the tube.
Helium stream replaced by pure oxygen for a brief period prior to sample introduction.
Sample instantaneously burned (flash combustion) followed by intense oxidation of tin capsule at 1800°C.
Resulting combustion gases passed over catalysts to ensure complete oxidation and absorption of halogens, sulfur and other interferences.
Excess oxygen removed as gases are swept through reduction tube containing copper at 650°C. Any oxides of nitrogen reduced to nitrogen gas.
Gases are then separated on chromatographic column into N, C, and H. These gases are quantitatively measured by a thermal conductivity detector.
Tapped density measurement:
Record the mass of the empty measuring cylinder. Fill the measuring cylinder to about 49.5mL and record the volume. Read the scale to the nearest 0.1 mL. Weigh the measuring cylinder filled with graphene and determine the mass of the graphene. Record the mass of graphene. Place the measuring cylinder into the Tap Density Tester and ensure it is held in place securely. Set the tap density tester for a time of 12 minutes and start the test. Once the sample has been tapped, record the volume of graphene in the measuring cylinder. Record to the nearest 0.1 mL.
5) The authors claim that the Pure GRAPHH 50 is redispersed in water after tip sonication. Pure Graphene ( or exfoliated graphite, which is what the authors have) is hydrophobic, it cannot be properly stabilized in aqueous solutions without the use of a surfactant. The authors should explain this, how it is possible? Authors should study XPS and Raman, the quality of the Pure GRAPH 50 after the sonication, it is possible that during the treatment, the surface of the flakes got oxidized. Therefore, having O and OH groups that may affect the physical properties of concrete and mortar.
In case of mortar preparation, the PureGRAPH® 50 was dispersed only in water using probe sonication. The lack of surfactant in this case resulted in the poor distribution of PureGRAPH® 50. During concrete preparation, the PureGRAPH® 50 was dispersed in water along with superplasticizer (MasterGlenium SKY 8100). This acted as the surfactant and improved the PureGRAPH® 50 dispersion.
This explanation was already mentioned in the manuscript Section 2.2, Line 238 and Section 3.1, Line 373-380. The images after the probe sonication were also given in the supplementary section – Figure SI-2.

Reviewer 2 Report
Regarding to the paper entitled “Enhancing Concrete and Mortar Properties and Durability Using Pristine Graphene Particles” that has been submitted to Coatings, here in follow some explanations in this matter are presented.
• Article innovation should be more explained clearly in comparison with other research works in this field.
• Present more details of test instruments in Section 2.3.
• Present and compare implementation costs of using Pristine Graphene Particles in Civil Engineering.
• Explain more interpretation and conclusion of results.
• Provide more references from recent years in relation to this topic.
• The English language of this manuscript should be improved.
Author Response
Dear Editor,
We are grateful for the constructive comments received from the reviewers, which helped us to further improve the quality and clarity of the manuscript. We appreciate the opportunity to revise this manuscript and have carefully evaluated and addressed all the comments and amended the manuscript accordingly. Manuscript ID: coatings-1996686 Below are our detailed responses to the reviewer’s comments point by point. The comments from the editor and reviewers are in black, responses from the authors are in blue, and revisions to the manuscript are in red. We would be happy to address any further comments that you or the reviewers might have.
Kind regards,
Dr. Guangming Jiang
School of Civil, Mining and Environmental Engineering,
University of Wollongong, Australia.
On behalf of all the authors
COMMENTS FROM THE EDITOR AND/OR REVIEWERS
Your manuscript has now been reviewed by experts in the field. Please find your manuscript with the referee reports. Please revise the manuscript according to the referees’ comments.
We appreciate all the constructive and insightful comments on the manuscript made by all reviewers and the editor. We carefully revised the manuscript by taking all these comments into consideration, as shown below.
Reviewer 2:
1) Article innovation should be more explained clearly in comparison with other research works in this field.
We agree with the reviewer that there are many other studies being done using various graphene products to enhance concrete or mortar properties. These studies were reviewed and discussed in the introduction section. While graphene materials are steadily making progress in terms of commercial adoption, with several graphene manufacturers across the world capable of producing industrially relevant quantities, there is still a lack of research on the use of industrially produced graphene to increase cementitious performance are still lacking. Therefore, this study systematically assessed the use of of industrially manufactured graphene with a focus on studying the dosage effects (0%-0.2% in concrete and 0%-0.15% in mortar) of small PRG (DV50 = 50 µm) particles on boosting the durability together with the mechanical and microstructural characteristics of both concrete and mortar. This study is systematic by integrating mechanical, microstructural and durability measurements, which are generally not reported in other studies. Especially, our research group mainly focus on the corrosion resistance and durability in urban wastewater systems. Therefore, this study is mainly designed to achieve good durability but also to investigate the supporting mechanisms by investigating the microstructural changes induced by the use of PRG. We have made the following changes to the manuscript to clearly distinguish this study from other published works.
Line 194:
Thus, the primary aim is to use industrially manufactured graphene nanoplatelets to achieve enhanced durability (resistance to acid induced deterioration) in corrosive environment like sewers and to investigate the positive changes of mechanical and microstructural properties that support the enhanced corrosion resistance.
2) Present more details of test instruments in Section 2.3.
Accepted and changes have been made.
Line 313:
Compressive strength tests were executed on the concrete and mortar in line with AS1012.9 and AS2350.11 respectively at an age of 28 days using 1800 kN Avery Compression Testing Machine.
Line 325:
For immersed absorption (Ai), to the nearest 0.1 g, the sample was weighed using a weighing balance (2200 g Scout General Portable Balance) and later was left in the oven (Thermoline Scientific Bench top- TO-SERIES) at 100 to 110 oC
3) Present and compare implementation costs of using Pristine Graphene Particles in Civil Engineering.
Thanks for the comments and this economic information is added to the manuscript.
Line 191-193:
Moreover, the cost of generating graphene in large amounts has significantly decreased because of industrial production. This is making it possible to include graphene into materials used on an industrial scale, such as concrete (Ho, 2020).
Line 208:
The estimated cost of graphene nanoplatelets used for this study is (USD) $20/kg.
4) Explain more interpretation and conclusion of results.
Accepted and changes have been made.
Line 564:
The high sulfate resistance may be also linked to the reduced AVPV values and subsequently the lower chemical diffusion rate in the mortar or concrete samples.
Line 607:
Thus, the reinforced concrete with graphene nanoplates meet the high requirement of durability for many new infrastructures to ensure adequate service life.
5) Provide more references from recent years in relation to this topic.
Accepted and changes have been made.
Line 72:
Nanofibers and nanotubes have been used as additives in cementitious composites and shown to increase mechanical properties (Gu et al., 2020; Konsta-Gdoutos et al., 2014; Pakharukov et al., 2019; Şimşek et al., 2022; Tyson et al., 2011).
Line 97:
Other researchers corroborated this by reporting that GO improved the strength and hardness of cement composites (Gu et al., 2020; Intarabut et al., 2022; Lv et al., 2014; Pakharukov et al., 2019).
Line 191:
Moreover, the cost of generating graphene in large amounts has significantly decreased because of industrial production. This is making it possible to include graphene into materials used on an industrial scale, such as concrete (Ho, 2020).
6) The English language of this manuscript should be improved.
The writing of this manuscript has been further improved by a native English speaker.

Reviewer 3 Report
Minor revision:
1.Redraw the new findings "Graphene reduced the AVPV of mortar and concrete by 12-19% at the optimal 19 dosage, possibly by reducing the number or pore size in the paste. The 0.2% and 0.15% graphene 20 reinforced concrete and mortar showed significant sulfate resistance, by reducing 62% and 60% of 21 extension respectively , after 16 weeks of exposure to sulfate solution"
2.Rewrite "Because of its flexibility, the main chain easily wraps around graphene 268 through hydrophobic and other intermolecular interactions. While the graphene disper- 269 sion in aqueous solutions is aided by the hydrophilic areas which hinder the graphene 270 agglomeration in water (WANG et al. 2012; Metaxa 2015)."
3.Recheck "Error! Reference source not found. and Table SI-7 depict the AVPV of the graphene 357 admixed concrete and mortar samples. The AVPV for the control concrete and mortar is 358 11.1% and 12.9% respectively. The addition of PureGRAPH® 50 at a concentration of 0.1% 359 decreased the AVPV by 11.7%, as compared to a decrease of 7.2% for 0.2% of graphene 360 against the control."
4.Recheck "Expansion because of 405 the exposure to sulfate solutions was found to reduce in a systematic order with pristine 406 graphene particle content of 0.07 % or more for both types of samples, with higher levels 407 in cement leading to reduced expansion. Attacking ions in the sulfate solution may pref- 408 erentially bind to graphene, rendering them ineffective on hydrate quality. (Sharma and 409 Arora 2018)."
Author Response
Dear Editor,
We are grateful for the constructive comments received from the reviewers, which helped us to further improve the quality and clarity of the manuscript. We appreciate the opportunity to revise this manuscript and have carefully evaluated and addressed all the comments and amended the manuscript accordingly. Manuscript ID: coatings-1996686 Below are our detailed responses to the reviewer’s comments point by point. The comments from the editor and reviewers are in black, responses from the authors are in blue, and revisions to the manuscript are in red. We would be happy to address any further comments that you or the reviewers might have.
Kind regards,
Dr. Guangming Jiang
School of Civil, Mining and Environmental Engineering,
University of Wollongong, Australia.
On behalf of all the authors
COMMENTS FROM THE EDITOR AND/OR REVIEWERS
Your manuscript has now been reviewed by experts in the field. Please find your manuscript with the referee reports. Please revise the manuscript according to the referees’ comments.
We appreciate all the constructive and insightful comments on the manuscript made by all reviewers and the editor. We carefully revised the manuscript by taking all these comments into consideration, as shown below.
Reviewer 3:
1) Redraw the new findings "Graphene reduced the AVPV of mortar and concrete by 12-19% at the optimal 19 dosage, possibly by reducing the number or pore size in the paste. The 0.2% and 0.15% graphene 20 reinforced concrete and mortar showed significant sulfate resistance, by reducing 62% and 60% of 21 extension respectively , after 16 weeks of exposure to sulfate solution".
Accepted and we draw a clearer conclusion from the findings.
Line 19:
Graphene reduced the AVPV of mortar by 11.7%, and concrete by 19.3% at the optimal dosages, likely by reducing the number or size of pores in the paste. The 0.2% and 0.15% graphene reinforced concrete and mortar showed significant sulfate resistance, by reducing 62% and 60% of extension respectively, after exposure to sulfate solution for 16 weeks.
2) Rewrite "Because of its flexibility, the main chain easily wraps around graphene 268 through hydrophobic and other intermolecular interactions. While the graphene disper- 269 sion in aqueous solutions is aided by the hydrophilic areas which hinder the graphene 270 agglomeration in water (WANG et al. 2012; Metaxa 2015)."
The sentence has been rephrased for better understanding.
Line 327-334:
The superplasticizer has a “comb-like” molecular structure consisting of an anionic adsorbing main chain and non-adsorbed non-ionic side chain (Wang et al. 2012; Q. Wang et al. 2020). Non-ionic surfactants can be employed to successfully disperse graphene in water at high concentrations. Anionic surfactants have been proven to be efficient at dispersing and stabilizing graphene particles in aqueous solutions even at low concentrations. Thus, the combined action of non-ionic and anionic components in the superplasticizer supports the ability to disperse the graphene in aqueous solution (Wang et al. 2012; Metaxa 2015).
3) Recheck "Error! Reference source not found. and Table SI-7 depict the AVPV of the graphene 357 admixed concrete and mortar samples. The AVPV for the control concrete and mortar is 358 11.1% and 12.9% respectively. The addition of PureGRAPH® 50 at a concentration of 0.1% 359 decreased the AVPV by 11.7%, as compared to a decrease of 7.2% for 0.2% of graphene 360 against the control."
Thanks for identifying the error and it has been modified.
Line 448:
Figure 3 and Table SI-7 depict the AVPV of the PureGRAPH® 50 admixed concrete and mortar samples. The AVPV for the control concrete and mortar is 11.1% and 12.9% respectively. The addition of PureGRAPH® 50 at a concentration of 0.1% decreased the AVPV by 11.7%, and for 0.2% concentration the AVPV reduced by 7.2% when compared to the control concrete.
4) Line 490-495: Recheck "Expansion because of the exposure to sulfate solutions was found to reduce in a systematic order with pristine graphene particle content of 0.07 % or more for both types of samples, with higher levels in cement leading to reduced expansion. Attacking ions in the sulfate solution may preferentially bind to graphene, rendering them ineffective on hydrate quality. (Sharma and 409 Arora 2018)."
Thanks for identifying the error and it has been modified.
Line 515:
The sulfate induced expansion was found to decrease consistently with pristine graphene particle content of 0.07 % or more for both types of samples. Sulfate ions may preferentially bind to graphene, rendering them ineffective on the quality of hydrates (Sharma and Arora 2018).

Round 2
Reviewer 1 Report
The authors called the compound “Graphene nanoplatelets” because it suits the material they have used.
They are still calling it graphene to something it is not graphene as it is more than one atom thick.
The definition of graphene accordingly to the IUPAC is "A single carbon layer of graphite", This is not the case, therefore, to use the term graphene is not acceptable.
Reviewer 2 Report
The paper has been corrected.